# Validation and Simultaneous Monitoring of 311 Pesticide Residues in Loamy Sand Agricultural Soils by LC-MS/MS and GC-MS/MS, Combined with QuEChERS-Based Extraction

**DOI:** 10.3390/molecules28114268

**Published:** 2023-05-23

**Authors:** Petros Tsiantas, Eleftheria Bempelou, Maria Doula, Helen Karasali

**Affiliations:** 1Laboratory of Chemical Control of Pesticides, Scientific Directorate of Pesticides’ Control and Phytopharmacy, Benaki Phytopathological Institute, 8 St. Delta Str., 145 61 Kifissia, Greece; p.tsiantas@bpi.gr; 2Laboratory of Non-Parasitic Diseases, Scientific Directorate of Phytopathology, Benaki Phytopathological Institute, 8 St. Delta Str., 145 61 Kifissia, Greece; m.doula@bpi.gr

**Keywords:** pesticide residues, soil, LC-MS/MS, GC-MS/MS, method validation, multi-residue method, matrix effect

## Abstract

Soil can be contaminated by pesticide residues through agricultural practices, by direct application or through spray-drift in cultivations. The dissipation of those chemicals in the soil may pose risks to the environment and human health. A simple and sensitive multi-residue analytical method was optimized and validated for the simultaneous determination of 311 active substances of pesticides in agricultural soils. The method involves sample preparation with QuEChERS-based extraction, and determination of the analytes with a combination of GC-MS/MS and LC-MS/MS techniques. Calibration plots were linear for both detectors over the range of five concentration levels, using matrix-matched calibration standards. The obtained recoveries from fortified-soil samples ranged from 70 to 119% and from 72.6 to 119% for GC-MS/MS and LC-MS/MS, respectively, while precision values were <20% in all cases. As regards the matrix effect (ME), signal suppression was observed in the liquid chromatography (LC)-amenable compounds, which was further estimated to be negligible. The gas chromatography (GC)-amenable compounds showed enhancement in the chromatographic response estimated as medium or strong ME. The calibrated limit of quantification (LOQ) value was 0.01 μg g^−1^ dry weight for most of the analytes, while the corresponding calculated limit of determination (LOD) value was 0.003 μg g^−1^ d.w. The proposed method was subsequently applied to agricultural soils from Greece, and positive determinations were obtained, among which were non-authorized compounds. The results indicate that the developed multi-residue method is fit for the purpose of analyzing low levels of pesticides in soil, according to EU requirements.

## 1. Introduction

The soil agro-environment can be contaminated with a variant of pollutants such as pesticides, antibiotics, microplastics, heavy metals and any other material which can potentially infect soil. Soil can be contaminated by pesticides through agricultural practices, by direct application or through spray-drift in crops. From soil, pesticide residues can be spread to aquifers, thereby affecting the quality of the agricultural crops and products. This in turn may be associated with several environmental impacts [1] and may affect the health of the consumers in the region [2]. Soil quality and crop production can also be negatively affected, putting at risk ecosystem services, nutrient cycling, enzyme activity, soil biota, and biodiversity [3,4].

Dissipation of chemical pesticides in soil depends on the characteristics of the soil, the nature of the chemical compounds, the cropping system, the irrigation pattern, and the ambient climatic conditions [5]. Soil characteristics vary continuously in terms of climate, parent materials, living organisms and crop management. Moreover, pesticides and/or biocides can be present in soil in the form of mixtures, which poses certain concerns because of their potential synergistic, additive, and/or antagonistic effects on non-target organisms [6].

Lack of knowledge on the fate and transference behavior of pesticide residues constitutes strong difficulties for soil and consequently land management, an issue that can be boosted further considering that the use of pesticides has been increasing in recent years for maintaining traditional agricultural practices [1]. Over the past decade, pesticides use surpassed 4 million tons per year worldwide [7]. The 2018 assessment from FAO shows that the use of agrochemicals is increasing in America and Asia, and the countries with the highest average use of pesticides since 1990 in relation to the arable land are, among others, Japan, the Republic of Korea, China, Israel, and Malta [8]. On a global level, among pesticides herbicides present the highest sales, followed by insecticides and fungicides [9].

The detection of those contaminants in soil is particularly challenging due to the low detection limits required, the complexion of the nature of the soil, and the difficulty in separating these compounds from interferences. Recent literature data revealed a high occurrence of mixtures of pesticide residues in arable fields treated with pesticides [10,11,12,13,14,15,16,17]. In regard to the preparation of soil samples prior to chromatographic analysis [18], the extraction process of soil and sediment samples is conducted through extraction with water/acetonitrile after the addition of the internal standard, while extraction with hexane has also been reported [19]. Packing the homogeneous soil mixture into a glass macro column with anhydrous sodium sulphate and silica gel and elution with hexane/acetone has been also reported [20], along with the analysis of soil samples by using a Soxhlet apparatus with hexane/acetone [21,22] and liquid/liquid extraction [23,24] have been also employed. In an effort to apply less time- and solvent-consuming techniques [25], the QuEChERS (quick, easy, cheap, effective, rugged and safe) method has been applied in a plethora of studies on soil, sediment, and sludge [12,15,26,27,28,29,30,31,32,33,34,35,36,37], and seems to be one of the most convenient, due to the high extraction yields that can be achieved [38]. Additionally, extraction of soil samples was achieved using the solid phase microextraction (SPME) fiber [39] οr by as solid phase microextraction (USE) [40], microwave-assisted extraction (MAE) [41] and microwave-assisted micellar extraction (MAME) [42]. The chromatographic techniques used in analytical methods for the identification and quantification of pesticide residues in soil are liquid chromatography coupled with a mass spectrometer detector such as UHPLC-MS/MS [18,43,44], LC-MS/MS [12,15,17,26,27,32,37,45,46], LC-QqTOF-MS [25], HPLC/MS [23], HPLC [24,29], and HPLC/DAD [28] and gas chromatography with different selective detectors such as GC/ECD [20,21,22,29], GC/NPD [47], GC/MS [19,22,48] and GC-MS/MS [26,27,30,34].

Many pesticides are characterized as persistent organic pollutants (POPs), which are toxic chemicals that negatively affect human health and the environment around the world. They persist for long periods of time in the environment and can be accumulated and transferred from one species to another through the food chain [49]. Amongst those compounds, there are organochlorines, being in wide usage across the world to control agricultural pests and vector-borne diseases [50,51]. Dichlorodiphenyltrichloroethane (DDT), hexachlorocyclohexane (HCH), endosulfan, aldrin, chlordane, dieldrin, endrin, heptachlor, mirex, hexachlorobenzene (HCB) and toxaphene, metoxychlor, and metolachlor are very stable compounds and their half-lives can range from a few months to several years; in some cases, decades [52]. The persistent organic pesticides found in the crops have likely been derived from soil, water, or air. Pesticide pollution in soil is the most important source of exposure [53]. High concentrations of organochlorine pesticides have been found not only in developing countries but also in industrialized ones, even though the use of many of these compounds has been restricted for least three decades. Based on the results of the published studies, organochlorine pesticides have been found in European, American, African, and Asian countries. OCPs have also been detected in many environmental compartments, along with mammals in the pristine regions of Arctic and Antarctic [54,55].

The challenges posed by emerging contaminants in soils are crucial, and require rigorous actions and collaboration. There is a need for data monitoring and risk assessment models, but also to raise awareness, new guidelines and governance models to deal with emerging contaminants in soils. Recent research trends have focused on different approaches to mitigate pesticide risks in soil, dealing with chemical remediation, containment or immobilization, and bioremediation [56,57,58,59] as potential strategies to improve the provision of ecosystem services and particularly to enhance rural livelihoods and development, contributing to poverty reduction. Monitoring of the pesticide residues in agricultural soils is the first and most fundamental step in the above procedures. Up until now, wide monitoring programs for pesticides are currently lacking for agricultural soils, with only minor exceptions [15]. Therefore, in the present study a multi-residue analytical method was developed and validated, capable of successfully determining the residues of 311 active compounds of pesticides from agricultural soils from Greece. The sample preparation of the method was based on the QuEChERS-approach, providing the opportunity to analyze all types of samples (plant or animal origin and soil) with the same method and using one extraction system. The determination of the compounds was performed by a combination of GC-MS/MS and LC-MS/MS chromatographic techniques. The proposed method was subsequently applied to 60 soil samples originating from agricultural areas in Greece, and their potential contamination with pesticide residues was examined.

## 2. Results

### 2.1. Determination of Compounds

#### 2.1.1. Optimization of MS/MS Parameters

Optimum quality parameters were achieved in both chromatographic systems (GC-MS/MS and LC-MS/MS) in the detection of all the target compounds. Through the completion of the optimization procedure, the optimized conditions for each substance were saved and applied in the determination.

#### 2.1.2. Optimization of the Extraction Efficiency

Investigation of the optimum solvent used during the sample preparation step occurred, following the extraction of spiked soil samples with four different extraction solvents for six replicates. As shown in Figure 1, in the case of the LC-determined analytes, the majority of the examined compounds gave average recoveries of above 70% for all the extraction solvent systems, with acetonitrile showing 100% of acceptable recoveries. In the case of GC, extraction with hexane resulted in low recoveries for almost all compounds, acetone (with or without formic acid) gave very good results, while 100% acceptable recoveries were obtained for acetonitrile. Therefore, and based on the above results, acetonitrile was proved to be the most efficient solvent for the extraction of the targeted compounds from soil samples in both LC-MS/MS and GC-MS/MS. Extraction with acetonitrile, the commonly used solvent in QuEChERS, will lead to the efficient determination of the analytes simultaneously in liquid and gas chromatography with an evaporation and reconstitution-of-solvent step.

### 2.2. Validation of the Analytical Method

Blank soil samples used in the validation of the developed analytical method showed no interferences in chromatograms of any of the examined analytes (Figure 2A and Figure 3A).

#### 2.2.1. Method Selectivity

The selectivity of the analytical method was proved by clearly labelled chromatograms of matrix-matched standard(s) at the lowest calibrated level, matrix blanks and samples fortified at the lowest fortification level, as shown in Figure 2 and Figure 3.

#### 2.2.2. Accuracy and Precision

Quantification of sample extracts during validation was performed by using single-point calibration standards and not by using a calibration curve. Based on this technique, the peak area of the sample solution was bracketed between two concentrations (not differing by more than 20%) instead of a single point.

Regarding the compounds determined by LC, all recovery values ranged from 72.59% to 118.73%, meeting the criterion of being between 70 and 120%, which is the acceptable interval according to SANTE/11312/2021 [60], with the only exception being two individual values of 64.64% (dimethomorph fortified at 0.01 μg g^−1^) and 69.63% (fenhexamid fortified at 0.01 μg g^−1^). Additionally, all the calculated relative standard deviation (%RSD) values met the requirement of <20%. Similarly, the obtained recoveries for the analytes determined by GC ranged from 70% to 119%, while the respective RSD (%) values were all <20%, demonstrating acceptable precision and accuracy. Results for both accuracy and precision for GC and LC amenable compounds are analytically presented in Appendix A, respectively.

#### 2.2.3. Linearity

The linearity of the method was assessed from the parameter of the calibration line using matrix-matched standards of five concentrations levels, to overcome matrix effects. With this approach, accurate sample quantification is achieved, since matrix effects may be responsible for systematic deviations of the analytical result from the ‘true’ value [61]. Both MS detectors gave linear response over the studied range of concentrations, and the least-squares linear regression analysis of the data provided excellent correlation for all compounds tested (r > 0.995).

#### 2.2.4. Limit of Quantification and Limit of Detection

Limit of quantification (LOQ) and limit of detection (LOD) are both important validation parameters, and were used to evaluate the sensitivity of the method. The LOQ was established as the lowest concentration level tested with acceptable accuracy and precision values. Therefore, for both chromatographic systems and for most of the compounds the LOQ was set at 0.01 μg g^−1^, while for 27 compounds it was set at 0.02 μg g^−1^ and for 13 compounds at 0.1 μg g^−1^ (Appendix A). LOD is related to LOQ with the equation 10 × LOD = 3 × LOQ, and therefore the calculated LODs of the method proposed were set to 0.003 μg g^−1^, 0.006 μg g^−1^ and 0.03 μg g^−1^, respectively.

#### 2.2.5. Matrix Effects (ME)

According to the results, ME depends on both the analyte and the chromatographic system. For example, 23.2% of the analytes determined with LC-[ESI-]-MS/MS have a positive ME in soil, meaning that components of the matrix induce ionization, whereas the usual phenomenon in LC-[ESI+]-MS/MS is signal suppression (the analyte molecules compared with other components of the matrix for protons available in the mobile phase), a case that was further confirmed in the present study, with 76.8% of analytes showing a negative ME. Additionally, as presented in Appendix A, only the analytes nuarimol and fluoxastrobin showed a medium ME, with most of the compounds showing negligible ME.

Opposite results were observed in the case of GC-MS/MS, where non-volatile matrix components accumulate in the inlet, masking active sites in the liner, which may increase the transfer of target pesticides to the detector, resulting in an enhancement of the chromatographic response. Only 7 compounds: di-allate-1, diazinon, fenoxycarb, hexachlorobenzene, pentachloroanisole, propachlor and trifluralin proved not to be affected by the matrix, whereas 152 analytes showed medium ME, and 31 analytes presented strong matrix effect. Nevertheless, in our study all calibrations were conducted using matrix-matched calibration standards, thus overcoming biased detections attributed to the substrate.

The method was found to be effective for the extraction of the tested compounds, and the above results indicate its efficiency for the determination of the target compounds from soil samples and ensure the accuracy of the residue analysis results (Figure 2 and Figure 3).

#### 2.2.6. Participation in Proficiency Testing Scheme

The proposed method was applied in the proficiency testing scheme PT-PAS-II organized by the Central Institute UKZUZ of the Czech Republic for supervision and testing in the Agriculture Department of proficiency testing programmes. The PT was about the determination of pesticides in agricultural soil, aiming for a comparison of the performances of participant laboratories in the field of pesticide analyses in soils. Each participant was provided with two soil test items and the soil blank. As reported by the organizer, all our obtained z-scores were well below the critical value of ±2 (−1.49 to 0.24) (Appendix A), confirming the analytical capability of our lab in implementing the proposed method.

### 2.3. Method Application

Sixty soil samples, originating from agricultural areas of Greece, were analyzed with the proposed method, so as to evaluate its applicability and to investigate the potential contamination of soil samples with a single or multiple residues of those specific pollutants. As presented in Figure 4, a percentage of 76% of the analyzed samples were proved to have detectable residues. Forty per cent (40%) of the samples with positive determinations were determined with GC-MS/MS and 26% with LC-MS/MS, while in 34% the results were confirmed by both chromatographic systems.

A total of 25 different compounds were determined, as presented in Table 1. The fungicide boscalid (GC-MS/MS) proved be the predominant, with 14 determinations, following by pendimethalin (fungicide, GC-MS/MS) with 9, methomyl (insecticide, LC-MS/MS) and penconazole (fungicide, LC-MS/MS) with 8 and p,p′-DDE (insecticide, GC-MS/MS) with 7 determinations. No metabolites or degradation products were detected.

As regards the detected concentrations in the analytes determined by GC-MS/MS, the maximum concentration observed was for p,p′-DDE, at 0.675 mg kg^−1^ d.w. The corresponding value in the case of LC-MS/MS was for thiophanate methyl, at 4.074 mg kg^−1^ d.w. It is acknowledged that, considering the data reported in Table 1, several fungicides (i.e., zoxamide, thiophanate methyl and dimethomorph) exist in sample extracts at concentrations above the highest calibration level. However, it must be mentioned that quantification was based on the single-point calibration matrix-matched standard. In each case, the matrix originated from the same sampling point and was previously analyzed so as to avoid the matrix effect, since the effects of different soil textures, pHs, organic matter and metal contents may influence the yield of the extraction [62,63].

Of these frequently detected compounds (Table 1), 52% were fungicides, 28% insecticides and 20% were herbicides. Moreover, most of them (56%) are approved for cultivation in the EU [64], while 44% (F: 36.4%, I:54.5%, H: 9.1%) are not.

Positive determinations of iprodione, pyraclostrobin, pyrimethanil, quintozene, procymidone, and chlorothalonil, were also determined in Yunnan Province [29]. Additionally, metolachlor, pendimethalin, azoxystrobin, carbendazim have also been determined in arable soils of the Czech Republic [65] and metolachlor, imidacloprid and tebuconazole in the depressed Pampas region of Argentina [18].

The organochlorine pesticides metolachlor, p,p′-DDD, p,p′-DDE and p,p′-DDT, were determined in the present study in Greek agricultural soils, in maximum concentrations of 0.021 mg kg^−1^, 0.023 mg kg^−1^, 0.675 mg kg^−1^ and 0.084 mg kg^−1^, respectively. Organochlorine residues have also been broadly distributed in Indian soil [66,67,68,69,70,71,72,73], in water and sediments from Dar as Salaam [74,75] and Zanzibar [76,77], in Kenyan estuaries [78] and in the Rufiji River Delta in Tanzania, where twenty-one organochlorine and organophosphate insecticides, and the herbicide thiobencarb occurred at quantifiable concentrations [79].

## 3. Materials and Methods

### 3.1. Studied Area and Sampling of Soil

Sampling of soils: A total of 60 (0–30 cm and 30–60 cm) soil samples from the selected agricultural areas of Greece were collected, to investigate the background soil contamination from pesticides and to verify their presence. All soil samples were collected with a soil auger and placed in plastic bags. Each sample consisted of 3 sub-samples within one farm site. Sub-samples were collected randomly and bulked together to form one composite sample. The samples were then sealed in clean polyethylene plastic bags and transported to the laboratory, where all samples were air-dried at room temperature in the dark, homogenized, and sieved through a 2 mm sieve. Finally, all samples were stored at −40 °C until extraction.

Soil properties such as texture (loamy sand, clay 6%, silt 14% and sand 80%), pH (7.15), and organic matter (1.4%) were determined using the Bouyoucos method, the soil paste and the liquid oxidation method, respectively in the respective soil. Organic carbon (0.83%) was determined by calculation. As expected, the soil organic carbon decreases with depth [80].

### 3.2. Selection of Analyzed Pesticides

The scope of the analytical method developed and applied in the present study comprised a wide range of pesticides that can potentially contaminate or be detected in soils from rural areas. In order to examine inappropriate or extensive past uses, a plethora of non-approved or long-banned active substances (e.g., organochlorines), in accordance with Regulation 1107/2009 are included within the objective of the study. Isomers, metabolites, or degradation products were also analyzed, especially in the cases that have been considered as relevant (toxicologically important compounds) during the evaluation of the active substance and the setting of the residue definition in soil. Finally, a list of 311 analytes were selected, comprising mainly the chemical classes of amides, carbamates, organophosphates, organochlorines, pyrethroids, sulfonylureas, strobylourines, triazines and dinitroanilines (Appendix A).

### 3.3. Analytical Methodology

#### 3.3.1. Chemicals and Solvents

The majority of the pesticide reference standards of the analytes that were determined with GC-MS/MS were included in a GC Multiresidue Pesticide Kit (Catalog. Number 32562) purchased from Restek, and were of ≥85% purity. Additionally, analytical standards of ametryn, azoxystrobin, boscalid, carboxin, difenoconazole, diflufenican, dimethenamid-P, dimethomorph, epoxiconazole, ethoprophos, famoxadone, fenoxaprop-P-ethyl, fenoxycarb, flufenacet, indoxacarb, kresoxim-methyl, mefenpyr-diethyl, metribuzin, napropamide, prometryn, quizalofop-P-ethyl and trifloxystrobin were purchased from Dr. Ehrenstorfer GmbH and Sigma-Aldrich, with a purity higher than 98%, except for Dimethenamid-P (94%).

Furthermore, the pesticide reference standards of the analytes determined with LC-MS/MS, were included in an LC Multiresidue Pesticide Kit (Catalog. Number 31971) purchased from Restek, and were of ≥85% purity.

Two mixes of salts obtained from Biotage were used for the QuEChERS extraction. The first mix consisted of 4 g Magnesium sulphate (MgSO_4_), 1 g Sodium chloride (NaCl), 1 g Trisodium citrate dehydrate and 0.5 g Disodium hydrogencitrate sesquihydrate (ISOLUTE^®^ QuEChERS EN 10 g/15 mL Extraction Tube) while the second one consisted of 150 mg Primary Secondary Amine (PSA) and 900 mg MgSO_4_ (ISOLUTE^®^ QuEChERS EN Fruit and Vegetables Clean-up Tube). Ammonium formate (HCOONH_4_) and formic acid (HCOOH) were purchased from Sigma Aldrich.

All the solvents, namely acetonitrile, methanol and water were purchased from Fisher Scientific (Loughborough, UK) and were of HPLC grade and pestipur for pesticide analysis. Acetone and hexane were obtained from Carlo Erba (Chau. du Vexin, France). All standard and sample solutions were filtered through a 0.22 μm hydrophilic nylon syringe filter (Membrane Solutions).

#### 3.3.2. Standard Solutions

GC—amenable pesticides: Each Multiresidue Pesticide Kit for GC-MS/MS analysis was separated into nine different ampules at a concentration of approximately 100 μg mL^−1^ in toluene. Each ampule was sonicated before use and diluted with acetone at a concentration of 10 μg mL^−1^ in a 10 mL volumetric flask (nine solutions). For those pesticides not included in the Multiresidue kit, individual standard stock solutions were prepared at 1000 μg mL^−1^ in acetone and stored at −40 °C. Composite working solutions at 10 μg mL^−1^ were also prepared in acetone and stored at −40 °C. Intermediate working solutions of 1 μg mL^−1^ were used for optimizing the mass spectrometer conditions.

LC—amenable pesticides: Each Multiresidue Pesticide Kit for LC-MS/MS analysis was separated into ten different ampules at a concentration of approximately 100 μg mL^−1^ in acetonitrile or methanol, and afterwards a composed mix of all ten solutions was prepared in a 20 mL volumetric flask and diluted with acetonitrile at a concentration of 1 μg mL^−1^.

Standard working solutions for the calibration curves of pesticides were prepared by serial dilution of the stock solution 1 μg mL^−1^ in acetonitrile. Matrix-matched calibration curves were prepared with the standard working-mix solutions, in soil extract. All pesticide solutions were stored in dark glass bottles at −40 °C. Stock solutions were kept for 18 months and working solutions for 3 days.

#### 3.3.3. Experimental Design

The extraction procedure of an analytical method is an important step, as it may crucially affect the quantification of pesticide residues. Among the factors that influence the yield of extraction are extraction time, procedure, and agitation, as well as the type of extraction solvent [81]. Therefore, aiming to achieve the best extraction efficiency, different extraction solvents were tested in both chromatographic systems. Regarding LC-MS/MS, blank soil samples (the same blank as for the validation procedure) were fortified and extracted with acetonitrile, acetonitrile + formic acid 1%, acetonitrile + acetic acid 1% and methanol + formic acid 1%. For each solvent, 6 replicates of samples fortified at 0.02 mg/kg were analyzed and the respective recoveries were calculated. Similarly, the GC-MS/MS fortification tests (n = 6) were applied with acetonitrile, acetone, acetone + formic acid 1% and hexane.

#### 3.3.4. Sample Preparation

##### Optimization of the Extraction Efficiency

To investigate the optimum extraction performance of each solvent, blank soil samples were spiked with all the tested analytes at 0.02 mg kg^−1^ and the samples were extracted following the QuEChERS approach, with the variation in the solvents. In total, four different solvent systems were tested per chromatographic system, and the obtained recoveries from each data set were calculated. The above experiment was conducted for six replicates per combination of analyte/solvent, providing a robust verification of the results.

Therefore, the extraction of pesticides from soil samples was carried out by the QuEChERS approach (BS EN 15662:2008). Concisely, soil samples were thawed and sieved, and a subsample of 5 ± 0.1 g was positioned into a 50 mL screw-cap polypropylene centrifuge tube, along with 10 mL of water (HPLC grade) and 10 mL of acetonitrile (HPLC grade), and shaken thoroughly for 1 min. Then, the first mix of salts (ISOLUTE^®^ QuEChERS EN 10 g/15 mL extraction tube) was added, and the mixture was agitated (end-over-end) for 1 min. After centrifugation at 3000 rpm for 5 min, 6 mL of the supernatant were collected in a separate centrifuge tube (15 mL screw cap) containing the second mix of salts (ISOLUTE^®^ QuEChERS EN Fruit and Vegetables Clean-up Tube). The mixture was shaken rigorously for 1 min and then centrifuged for 5 min at 3000 rpm. Finally, an aliquot of supernatant extract was filtered through 0.20 μm and analyzed by GC-MS/MS or LC-MS/MS.

#### 3.3.5. Determination of Compounds—Instrumentation

##### Analysis of Samples with Gas Chromatography

A SHIMADZU GC-2030 gas chromatograph (Shimadzu, Kyoto, Japan) coupled with a GCMS-TQ8040 NX mass spectrometer detector, equipped with a SHIMADZU AOC-6000 Autosampler was used. The GC was equipped with a split/splitless injector (injection system set at 250 °C, using a 3.4 mm splitless deactivated liner with glass wool (SHIMADZU)), operated in the splitless mode. The analytical capillary column was a MEGA-5MS capillary column (30 m × 0.25 mm; i.d. × 0.25 mm film thickness). Injection volume was set at 1 μL. The oven temperature was programmed to be maintained at 50 °C for 1 min, increased to 125 °C at a rate of 25 °C min^−1^, then increased to 300 °C at a rate of 10 °C min^−1^ and held at this temperature for 15 min. The carrier gas was He, with a flow rate of 1.69 mL min^−1^. The detector was operating in electron ionization mode, with electron energy of 70 eV. The temperature of the interface and the ion source were 250 °C and 230 °C, respectively. The total GC run time was 36.50 min. For instrument control and data acquisition and processing, the GCMS solution software (Version 4.50) was used. Confirmation of the determined analytes was based on the criteria of retention time and the ion abundance of 3 selected ions. The multiple reaction monitoring (MRM) mode was used for quantitative analysis. The quantification and qualification ions in the pesticides are listed in Appendix A.

##### Analysis of Samples with Liquid Chromatography

The liquid chromatography (LC) system used was a SHIMADZU (LC-MS 8050). The chromatographic separation was achieved using a reversed-phase Eclipse XDB-C18 column (150 mm × 2.1 mm × 3.5 μm particle size) (Agilent), operated at 40 °C. The flow rate was set at 0.25 mL min^−1^. The mobile phases consisted of two solvents, (i) 80% H_2_O, 20% MEOH, 5 mM HCOONH_4_ & 0.1% HCOOH (Solvent A) and (ii) ΜΕOH, 5 mM HCCONH_4_ and 0.1% HCOOH (Solvent B). A binary gradient using mobile phases A and B was programmed as follows: 0–2 min, 100% A, 2–12 min, 50% B, 12–30 min, 100% B, 30–37 min 100% B and 37.01–45 min, 100% A.

To avoid carryover, the autosampler was purged with acetonitrile by the operation program of the instrument, before and after each injection of the samples. Total run time was achieved at 45 min. The mass spectrometer used was LCMS 8050, equipped with an electrospray ionization (EI) interface. The ESI–MS interface was operated both in the positive- and the negative-ion detection mode, at 200 °C. The ESI source conditions were: desolvation temperature 355 °C, desolvation line 250 °C, heat block temperature 400 °C, conversion dynode 6 kV, nebulizing gas flow 2 L min^−1^, heating gas flow 10 L min^−1^ and drying gas flow 10 L min^−1^ (both nebulizer and drying gas were high-purity nitrogen, produced by a high-purity generator). MS–MS experiments were carried out with argon (purity 99.9%) at a pressure of approximately 1.5 mTorr in the collision cell. The MRM mode (Appendix A) was applied for the detection and quantification of the active ingredient under study. The source parameters were optimized using the Lab Solution 5.99 SP2 software.

##### Optimization of MS/MS Parameters

During the development of the analytical method in the GC-MS/MS system, satisfactory quality parameters were achieved in the detection of the target compounds. In the case of the LC-MS/MS system, optimization of the chromatographic analysis parameters was deemed necessary. Specifically, the optimized parameters related to the accuracy of the main ions (quantification ion) and the fragmentation ions (identification ion), as well as the corresponding energies Q1 Pre Bias (V), Collision Energy (CE) and Q3 Pre Bias (V). Optimization occurred using the software of the instrument, and concerned all the substances that were validated separately. The procedure applied was flow injection analysis (FIA), without a chromatographic column, using the elution solvents in a ratio of 50/50, with a flow of 0.2 mL min^−1^. The analysis time of each injection was 1 min, and the injected concentration was 100 ppb. The analyte passed directly to the ionization source and, based on the automated program, the molecular ions and their energies were adjusted, according to the greater sensitivity obtained. By the completion of the procedure, the optimized conditions for each substance were saved.

#### 3.3.6. Validation Procedure

The method was validated by assessing accuracy and precision (based on the results of the recovery experiments), linearity, sensitivity, and the matrix effect. A soil sample free from pesticide contamination was taken from a field close to the studied area, without previous history of pesticide use. This substrate was previously analyzed to ensure that it did not contain any of the studied compounds, therefore avoiding interferences, and it was used as a blank soil sample.

The accuracy (percentage recoveries) and precision (% relative standard deviation) of the method was confirmed by measuring recoveries from spiked blank soil at three concentrations levels, i.e., at 0.01, 0.02 and 0.1 μg g^−1^ for the GC amenable compounds and at four fortification levels, i.e., at 0.01, 0.02, 0.05 and 0.1 μg g^−1^ for the LC amenable compounds, and presented in Appendix A, respectively. The blank soil used for the determination of the recovery of the method was previously analyzed to certify that it did not contain any of the studied compounds.

The spiking procedure was treated as follows: 5 g of the blank soil sample was placed into a 50 mL screw-cap polypropylene centrifuge tube, along with the appropriate quantity of the standard mixture of the studied pesticides in acetone for GC-amenable compounds and in acetonitrile for LC-amenable compounds. Then, it was homogenized by rigorous hand shaking for better analyte distribution and was left to stand for 1 h prior to extraction. Finally, the spiked samples were extracted in the same way as described in the sample preparation (Section 3.3.4). Samples were quantified with matrix-matched standards. All experiments were executed six times and the relative standard deviation (RSD %) was considered. Afterwards, the values obtained were used for the assessment of the precision of the extraction method. Mean recovery data and relative standard deviations (RSDs) expressing the precision of the extraction method are given in Appendix A for GC- and LC-amenable compounds, respectively.

Linearity was estimated at 5 concentrations levels, as 0.01, 0.02, 0.05, 0.1 and 0.2 μg mL^−1^ for the LC-amenable compounds and at 0.005, 0.01, 0.02, 0.05 and 0.1 μg mL^−1^ for the GC-amenable compounds. Four replicates were performed for each calibration point. Calibration curves were constructed with working standards as well as matrix-matched standards.

Sensitivity was estimated with the validated limit of quantification (LOQ) set to the lowest concentration tested (expressed in μg g^−1^ dry weight) for which a recovery in the 70–120% range could be obtained, with a corresponding RSD ≤ 20% and signal-to-noise (S/N) ratio higher than 10, according to the guidance document on residue analytical methods.

For matrix effect experiments, 5 calibration curve- levels were prepared in acetone for GC-amenable compounds and in acetonitrile for LC-amenable compounds and the soil matrix in duplicate. The soil matrix was extracted in the same way as described in the sample preparation. Matrix effects are often caused by the alteration of the ionization efficiency of target analytes in the presence of co-eluting compounds in the same matrix. The matrix effect can be observed either as a loss in response (ion suppression) or as an increase in response (ion enhancement). Both the ion suppression and enhancement dramatically affect the analytical performance of a method [82]. Matrix effects can be calculated by comparing the slope of the calibration curves, where the analytical standards are prepared either in pure solvent (b_solvent_) or in the matrix extract (b_matrix_) and % ME can be classified in three categories: no matrix effects (|ΜΕ| < 20%), medium matrix effects (20% < |ΜΕ| < 50%) and strong matrix effects (|ΜΕ| > 50%), based on the equation below [83].
% ME *=* 100 × (b_matrix_ − b_solvent_)/b_solvent_

## 4. Conclusions

In the present study a QuECheRS method for the determination of pesticide residues (active substances, isomers, metabolites, and degradation products) in agricultural soils was developed and validated, using GC-MS/MS and LC-MS/MS techniques. The obtained results were acceptable in both chromatographic systems, showing good separation, sensitivity, linearity, precision, and accuracy. Validation criteria were proved to satisfy the European guideline SANTE/11312/2021 [60] and the method was found to be effective for the extraction of the tested compounds. The proposed method was further assessed by the analysis of real samples from rural regions of Greece. Our findings mainly comprised insecticides, fungicides and herbicides, with the most frequently detected compounds being pp′-DDE, boscalid and pendimethalin, respectively. Additionally, an important number of detected pesticides have either been long banned or are non-authorized for use in agricultural soils in the EU. Based on the above, the method can be proposed as suitable for the quantitative and qualitative analysis of pesticide residues in agricultural soils, at low levels. However, it is highly recommended that, due to the complexity and the specificity of the matrix, during the monitoring of soil samples the choice of the matrix used in matrix-matched standards should resemble the type of soil analysis.

## Figures and Tables

**Figure 1 molecules-28-04268-f001:**
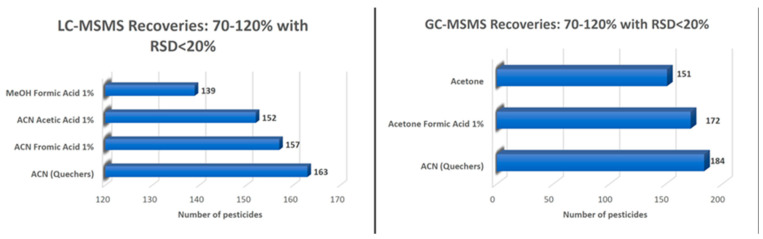
Influence of different extraction solvent systems on the recoveries of all targeted compounds after the fortification of soil samples at 0.02 mg kg^−1^ for six replicates determined for the LC-MS/MS and the GC-MS/MS chromatographic system, respectively.

**Figure 2 molecules-28-04268-f002:**
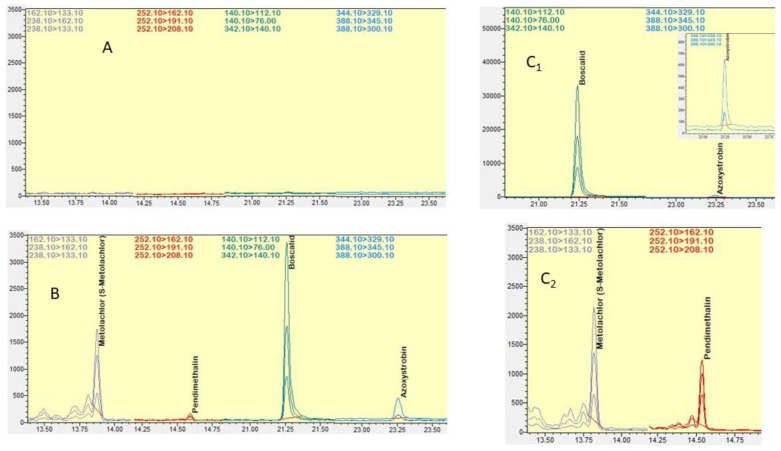
Detection of S-metolachlor, pendimethalin, boscalid and azoxystrobin in the GC/MS/MS chromatographic system: (**A**) blank soil sample, (**B**) in fortified soil sample, and (**C_1_**,**C_2_**) in agricultural soil samples.

**Figure 3 molecules-28-04268-f003:**
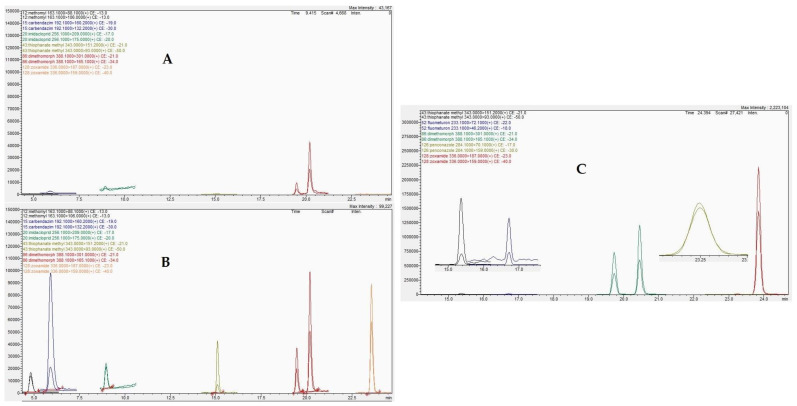
Detection of carbendazim, penconazole, thiophanate methyl, fluometuron, dimethomorph and zoxamide in the LC/MS/MS chromatographic system: (**A**) in blank soil sample (**B**) in fortified soil sample and (**C**) in agricultural soil samples.

**Figure 4 molecules-28-04268-f004:**
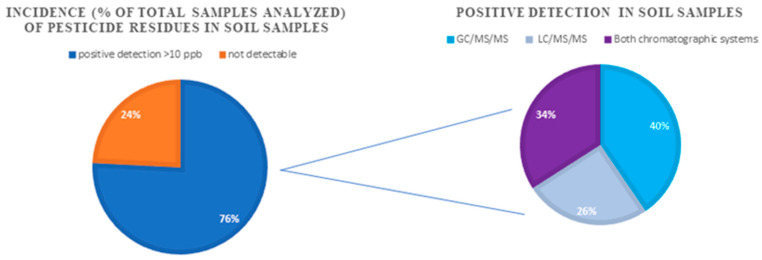
Incidence (% of total samples analyzed) of pesticide residues in soil samples.

**Table 1 molecules-28-04268-t001:** Type, frequency, and maximum concentration determined authorization for application in crops cultivated in the EU.

Pesticide	Type	Frequency (%)	Max Concentration (mg/kg)	Approval (EC, 2022)
Azoxystrobin	F	16	0.047	Yes
Boscalid	F	56	0.11	Yes
Carbendazim	F	4	0.037	No
Carfentrazon ethyl	H	8	0.03	Yes
Clothianidin	I	4	0.01	No
Dimethomorph	F	20	1.225	Yes
Epoxiconazole	F	8	0.01	No
Fludioxonil	F	4	0.01	Yes
Fluometuron	H	12	0.016	Yes
Imidacloprid	I	20	0.024	No
Metconazole	F	8	0.01	Yes
Methomyl	I	32	0.147	No
Metolachlor	H	4	0.021	No
Metribuzin	H	8	0.015	Yes
Myclobutanil	F	16	0.02	No
p,p′-DDD	I	12	0.023	No
p,p′-DDE	I	28	0.675	No
p,p′-DDT	I	16	0.084	No
Penconazole	F	32	0.032	Yes
Pendimethalin	H	36	0.341	Yes
Pyraclostrobin	F	4	0.014	Yes
Spinosyn A	I	4	0.01	Yes
Tebuconazole	F	8	0.025	Yes
Thiophanate methyl	F	12	4.074	No
Zoxamide	F	20	1.278	Yes

## Data Availability

Not applicable.

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
