# Peer review of "Validation and Simultaneous Monitoring of 311 Pesticide Residues in Loamy Sand Agricultural Soils by LC-MS/MS and GC-MS/MS, Combined with QuEChERS-Based Extraction"

_molecules, 2023, doi:10.3390/molecules28114268_

Round 1

Reviewer 1 Report

The Ms ‘Validation and simultaneously monitoring of 311 pesticide residues in loamy sand agricultural soils by LC-MS/MS and GC-MS/MS combined with QuEChERS-based extraction’ clearly depicted the concept and very well described the results of validation and their correlation with the real soil samples. Although the manuscript is written in a good manner yet some reservations are observed from my side which are as follows:

1. There is a great need to improve the English language in whole Ms. Many syntax and grammatical errors are observed. For eg. sentences in line 27-28; 63-66; 250-252 and many more like this

needs to be improved.
2. Section 2.3. How many samples were detected with residues above EU MRLs, needs to be supplemented in this section and Table 1 along with the MRLs value. 
3. Method optimization parameters as described in Results and Discussion section 2.1 should be added to Material and methods section. 
4. How could author able to validate all the selected 311 pesticide residues in acceptable recovery range and no pesticide detected with lower/high recovery or recovery outside the acceptable range of 70-120%. Explain and justify. 
5. How the author was able to schedule the PT program soon after the validation studies. Were the validation performed long before and during due course of time PT was performed, thereafter the validation along with PT results were compiled and sent for publication? Is this so? Or how the study was designed?

6. What is the novelty of the study as many validation and development regarding the pesticide and soil combination were published previously?

Author Response

Response to the Reviewer 1

For the revision of the paper

“Validation and simultaneously monitoring of 311 pesticide residues in loamy sand agricultural soils by LC-MS/MS and GC-MS/MS combined with QuEChERS-based extraction” (Molecules-2335381)

By P. Tsiantas et al.

Reviewer: The Ms ‘Validation and simultaneously monitoring of 311 pesticide residues in loamy sand agricultural soils by LC-MS/MS and GC-MS/MS combined with QuEChERS-based extraction’ clearly depicted the concept and very well described the results of validation and their correlation with the real soil samples. Although the manuscript is written in a good manner, yet some reservations are observed from my side which are as follows:

Answer: First of all, we sincerely thank the reviewer for his/her time and efforts to review towards the improvement of our work. His/her comments have been very valuable for us. Every comment/point the reviewer raised in the text, it is indicated with track changes in the revised draft.

Reviewer: There is a great need to improve the English language in whole Ms. Many syntax and grammatical errors are observed. For eg. sentences in line 27-28; 63-66; 250-252 and many more like this need to be improved.

Answer: Thank you for your comment. The manuscript has been revised by a native speaker. Sentences in lines: 27-28; 63-66; 250-252, have been revised properly.

Reviewer: Section 2.3. How many samples were detected with residues above EU MRLs, needs to be supplemented in this section and Table 1 along with the MRLs value. 

Answer: Thank you for your comment. There are no EU MRLs in soil. In the case of soils there are the ‘Soil Screening Values -SSVs’ which are generic standards adopted in some EU countries to regulate the management of contaminated sites. SSVs are in the form of concentration thresholds (like MRLs) above of which certain actions are enforced or recommended, however there is a large variability in SSVs values adopted within EU countries. The EU soil health law, that will propose unique thresholds within EU will be issued by the end of current year (2023). Thus, there are no unique values to be used until now.

Reviewer:  Method optimization Materials as described in Results and Discussion section 2.1 should be added to Material and methods section. 

Answer: Thank you for your comment. Section 2.1 Optimization of experimental parameters has been removed to Materials and Methods.

Reviewer:  How could author able to validate all the selected 311 pesticide residues in acceptable recovery range and no pesticide detected with lower/high recovery or recovery outside the acceptable range of 70-120%. Explain and justify. 

Answer: Thank you for your comment. It has been reported the results only for the pesticides that have acceptable recoveries. Active substances with not acceptable recoveries have been excluded from the method. Approximately 20 pesticides with recoveries outside the acceptable range of 70%-120% have been excluded from the method. The method for the 311 pesticides presents acceptable recoveries considering the results from the number of replicates (6 replicates) per fortification level.

Reviewer:  How the author was able to schedule the PT program soon after the validation studies. Was the validation performed long before and during due course of time PT was performed, thereafter the validation along with PT results were compiled and sent for publication? Is this so? Or how the study was designed?

Answer: Thank you for your comment. The PT was organized by UKZUZ, NRL, Department of Proficiency Testing Programmes (OdMPZ), of Czech Republic in 2022. The method has been validated before the PT, as our laboratory is the official lab for testing soil samples for pesticide residues in Greece. The PT is mentioned to provide evidence that the proposed method provides acceptable results that have been evaluated by an external body.

The study was planned as follows:

The proposed method has been developed and validated for analyzing official samples. For accreditation purposes, participation in a PT is obligatory. That was one of the reasons for participating in the PT. We decided to publish the method, due to the large number of analytes determined with a fast sample preparation method (QuEChERS).

Reviewer:   What is the novelty of the study as many validation and development regarding the pesticide and soil combination were published previously?

Answer: Thank you for your comment. The significance and the novelty of this work is to provide a method for the determination of pesticide residues in soil with a fast and efficient sample preparation step and a broad spectrum of pesticides (311). The method has been validated for 311 different pesticides, while the already published methods had been validated for a much lower number of pesticides (the maximum number is 225) based on the review article: González-Curbelo MÁ, Varela-Martínez DA, Riaño-Herrera DA. Pesticide-Residue Analysis in Soils by the QuEChERS Method: A Review. Molecules. 2022 Jul 5;27(13):4323. doi: 10.3390/molecules27134323. PMID: 35807567; PMCID: PMC9268078.

Reviewer 2 Report

The author established an analytical method for simultaneously determination of 311 pesticide in soils by LC-MS/MS and GC-MS/MS combined with QuEChERS-based extraction. This study provides important information for the residue determination and risk evaluation of pesticides on environmental safety. Overall, this paper could be published after moderate revision which were listed as follows:

1. Line 84-91: The residue distribution of some persistent pesticides should be described in this section.

2. Line 134: To verify the extraction efficiency of different organic reagents, what is the basis for selecting 0.02mg/kg concentration?

3. Line 143: According to the authors description, the extraction efficiency of acetone is also very good, so why only choose acetonitrile as the extraction reagent.

4. The resolution of the picture is poor.

5. Section 2.2.3. How to deal with the sample detection concentration exceeding the linear range.

6. Line 252: The maximum residual concentration of thiophanate-methyl is 4.074mg/kg, and the maximum concentration verified by the experimental method in this paper is 0.1ug/g. Please supplement a verification data with high concentration to prove that this method is also applicable at high concentration.

Author Response

Response to the Reviewer 1

For the revision of the paper

“Validation and simultaneously monitoring of 311 pesticide residues in loamy sand agricultural soils by LC-MS/MS and GC-MS/MS combined with QuEChERS-based extraction” (Molecules-2335381)

By P. Tsiantas et al.

Reviewer: The author established an analytical method for simultaneously determination of 311 pesticide in soils by LC-MS/MS and GC-MS/MS combined with QuEChERS-based extraction. This study provides important information for the residue determination and risk evaluation of pesticides on environmental safety. Overall, this paper could be published after moderate revision which were listed as follows:

Answer: First, we sincerely thank the reviewer for his/her time and efforts to review towards the improvement of our work. His/her comments have been very valuable for us. Every comment/point the reviewer raised in the text, is indicated with track changes in the revised draft.

Reviewer: Line 84-91: The residue distribution of some persistent pesticides should be described in this section.

Answer: We thank the reviewer for this valuable comment. A text describing the distribution of persistent organochlorine pesticides has been added to the text (after line 91). Please see also the text below.

High concentrations of organochlorine pesticides have been found not only in developing countries but also in industrialized ones even though the use of many of these compounds have been restricted at least three decades ago. Based on the results of the published studies organochlorine pesticides have been found in European, American, African, and Asian countries. OCPs have also been detected in many environmental compartments along with mammals in the pristine regions of Arctic and Antarctic (Tanabe, S., Hidaka, H. and Tatsukawa, R., 1983a. PCBs and chlorinated hydrocarbon pesticides in Antarctic atmosphere and hydrosphere. Chemosphere, 12: 227-288) and Tzanetou, E.N.; Karasali, H. A Comprehensive Review of Organochlorine Pesticide Monitoring in Agricultural Soils: The Silent Threat of a Conventional Agricultural Past. Agriculture 2022, 12, 728. https://doi.org/10.3390/agriculture12050728.

Reviewer: Line 134: To verify the extraction efficiency of different organic reagents, what is the basis for selecting 0.02mg/kg concentration?

Answer: Thank you for your comment. The concentration of 0.02mg/kg was selected aiming to ensure the successful determination of the majority of the compounds.

Reviewer:  Line 143: According to the authors description, the extraction efficiency of acetone is also very good, so why only choose acetonitrile as the extraction reagent.

Answer: Thank you for your comment. Indeed, based on the obtained results extraction with acetone gave also acceptable recoveries in the GC-MS/MS. However, we chose acetonitrile as the extraction solvent, which also showed successful performance, aiming to have one sample preparation step for both GC and LC amenable compounds.

Reviewer:   The resolution of the picture is poor.

Answer: Thank you for your comment. The figure has been replaced by Table 1, in which the assessment of matrix effect is presented for all analytes.

Reviewer:  Section 2.2.3. How to deal with the sample detection concentration exceeding the linear range.

Answer: Thank you for your comment. It is acknowledged that in the soil samples analyzed there were concentrations above the highest calibration level. In those cases, dilution of the sample solution was carried out as to join linearity range and the dilution factor was considered during calculations.  

Reviewer:   Line 252: The maximum residual concentration of thiophanate-methyl is 4.074mg/kg, and the maximum concentration verified by the experimental method in this paper is 0.1ug/g. Please supplement verification data with high concentration to prove that this method is also applicable at high concentration.

Answer: Thank you for your comment. As stated in the previous comment, in the cases (as thiophanate-methyl) in which indications of higher concentration were occurred, succeeding dilution of the sample solution were taken place as to satisfy linearity range.

Reviewer 3 Report

The paper is interesting for the publication anyway some revisions are need:

1) 2.2.3 linearity indicate the number of replicate for each point.

2) 2.2.5 The figure 4 is not readable and the authors should give the values with tables.

3)2.2.6 the information should be integrate with the substances analyzed and finding and with the values of each z-score/substance.

4) 3.1 the authors should integrate with the information on the origin of soil used as blank.

5) 3.3.1 indicate the purity for each standard used.

6) 3.3.7 delete the paragraph or implement the informations for example give the information on the acceptability criteria to confirm the substances.

7) References - the authors should follow the instuction of the journal for refernces.

Author Response

Response to the Reviewer 3

For the revision of the paper

“Validation and simultaneously monitoring of 311 pesticide residues in loamy sand agricultural soils by LC-MS/MS and GC-MS/MS combined with QuEChERS-based extraction” (Molecules-2335381)

By P. Tsiantas et al.

Reviewer: The paper is interesting for the publication anyway some revisions are need:

Answer: First of all, we sincerely thank the reviewer for his/her time and efforts to review towards the improvement of our work. His/her comments have been very valuable for us. Every comment/point the reviewer raised in the text, it is indicated with track changes in the revised draft.

Reviewer: 2.2.3 linearity indicates the number of replicates for each point.

Answer: Four replicates had been performed for each calibration point in the linearity. The number of replicates will be added to the manuscript.

Reviewer: 2.2.5 Figure 4 is not readable, and the authors should give the values with tables.

Answer: Thank you for your comment. The figure has been replaced by Table 1, in which the assessment of matrix effect is presented for all analytes.

Reviewer: 2.2.6 the information should be integrated with the substances analyzed and finding and with the values of each z-score/substance.

Answer: Z-scores/substance are presented in the Table below. The achieved z-scores ranges from 0.00 to -1.49. Table S3 with the z-scores should be added as supplementary materials.

Table S3. Z-score results after the participation of the validated method in the Proficiency Test for the determination of pesticide residues in soil. (Laboratory code: 9973)

Analyte

z-scores

PT-PASS-2022-S1

PT-PASS-2022-S2

Azoxystrobin

-1.49

Carbendazim

-0.50

-0.71

Clomazone

<LOQ

-0.76

Cyproconazole

<LOQ

<LOQ

Difenoconazole

<LOQ

<LOQ

Diflufenican

-0.58

0.00

Dimethachlor

<LOQ

-0.79

Epoxiconazole

-1.04

-0.65

Fenpropimorph

<LOQ

<LOQ

Flufenacet

<LOQ

<LOQ

Fluoxastrobin

<LOQ

Flusilazole

-0.43

0.62

Chlorotoluron

<LOQ

<LOQ

Chlorpyrifos

<LOQ

<LOQ

Imidacloprid

<LOQ

<LOQ

Linuron

<LOQ

Metconazole

<LOQ

-0.08

Nanpropamide

<LOQ

Picoxystrobin

<LOQ

Prochloraz

<LOQ

-1.10

Prometryn

<LOQ

Propiconazole

<LOQ

Spiroxamine

0.24

0.14

Tebuconazole

-0.68

-0.26

Terbuthylazine

<LOQ

<LOQ

Thiacloprid

<LOQ

<LOQ

Triadimenol

<LOQ

<LOQ

Trifloxystrobin

<LOQ

<LOQ

Atrazine

<LOQ

<LOQ

Metazachlor

<LOQ

<LOQ

Reviewer:  3.1 the authors should integrate with the information on the origin of soil used as blank.

Answer: We thank the reviewer for his/her valuable comment. The following text was added to the manuscript for better understanding:

A soil sample free from pesticide contamination was taken from a field close to the studied area, without previous history on pesticide use. This substrate was previously analyzed to ensure that it did not contain any of the studied compounds and therefore avoid interferences, and used as blank soil sample.

Reviewer:  3.3.1 indicate the purity for each standard used.

Answer:  Thank you for your comment. The purity of all validated analytes has been added in a new column the supplementary tables S1 and S2, for the GC and the LC amenable compounds, respectively.

Reviewer:   3.3.7 delete the paragraph or implement the information, for example give the information on the acceptability criteria to confirm the substances.

Answer: Thank you for your comment. The paragraph has been deleted and a relative sentence has been added in the Conclusion section.

Reviewer: References - the authors should follow the instruction of the journal for references.

Answer: Thank you for your comment. References have been revised in accordance with the instructions of the journal.

Round 2

Reviewer 1 Report

Reviewer’s comment

The authors have given responses to the comments but still some corrections are needed.

1.     English language for sentences in line 27-28 and 250-252 are yet to be corrected. The rest of corrected sentences are okay.

2.     As pointed out previously, method optimization parameters needs to be added to Material and methods section and the results of method optimization should be added to the Results and Discussion section.

The authors have removed the entire section as such from Results and Discussion and moved to Material and Methods.

3.     Table 1 Matrix effects assessed in LC-MS/MS and GC-MS/MS systems for all analytes. This table in my opinion is not required in the main Ms and can be added to supplementary material.

4.     Abbreviations must be accompanied by full forms while using it at first instance, including in the abstract.

Author Response

Response to the Reviewer 2

For the revision of the paper

“Validation and simultaneously monitoring of 311 pesticide residues in loamy sand agricultural soils by LC-MS/MS and GC-MS/MS combined with QuEChERS-based extraction” (Molecules-2335381)

By P. Tsiantas et al.

Reviewer: The authors have given responses to the comments but still some corrections are needed.

Answer: First of all, we sincerely thank the reviewer for his/her time and efforts to review towards the revised version of our work. His/her comments have been very valuable for us. Every comment/point the reviewer raised in the text, it is indicated with track changes in the revised draft.

Reviewer: English language for sentences in line 27-28 and 250-252 are yet to be corrected. The rest of corrected sentences are okay.

Answer: Thank you for your comment. Both sentences have been edited and corrected as recommended.

Reviewer: As pointed out previously, method optimization parameters needs to be added to Material and methods section and the results of method optimization should be added to the Results and Discussion section.

The authors have removed the entire section as such from Results and Discussion and moved to Material and Methods.

Answer: Thank you for your comment. The experimental of the optimization of the method has been described in the “Materials and Methods” section and the corresponding obtained results in the “Results” section.

Reviewer:  Table 1 Matrix effects assessed in LC-MS/MS and GC-MS/MS systems for all analytes. This table in my opinion is not required in the main Ms and can be added to supplementary material.

Answer: Thank you for your comment. Table 1 has been transferred to supplementary material.

Reviewer:  Abbreviations must be accompanied by full forms while using it at first instance, including in the abstract.

Answer: Thank you for your comment. Abbreviations must be accompanied by full forms.